# The Need for a Data Ecosystem for Youth Mental Health in The Netherlands

**DOI:** 10.3390/ijerph191811499

**Published:** 2022-09-13

**Authors:** Marloes Kleinjan, Danielle E. M. C. Jansen, Maartje van den Essenburg

**Affiliations:** 1Trimbos Institute, Netherlands Institute of Mental Health and Addiction, P.O. Box 725, 3500 AS Utrecht, The Netherlands; 2Interdisciplinary Social Science, Youth Studies, Utrecht University, P.O. Box 80140, 3508 TC Utrecht, The Netherlands; 3Department of General Practice & Elderly Care Medicine, University Medical Centre Groningen, Hanzeplein 1, 9712 CP Groningen, The Netherlands; 4Department of Sociology and Interuniversity Centre for Social Science Theory and Methodology (ICS), University of Groningen, Grote Rozenstraat 31, 9712 TG Groningen, The Netherlands; 5Accare, University Centre for Child and Adolescent Psychiatry, Lübeckweg 2, 9723 HE Groningen, The Netherlands

**Keywords:** mental health, data, infrastructure, youth, population data, registration data

## Abstract

The Netherlands is missing nationally representative data on child and adolescent mental health, e.g., on prevalence, course, and consequences of psychological disorders and mental health care utilization. Researchers and policy makers also lack a basic data infrastructure that is necessary to provide timely and reliable data crucial for benchmarking and informed decision making. In this article, we describe the necessity for a clear and well-organized overview of data on youth mental health and mental health care. We look back on three key moments in time to illustrate the breadth of the desire for data. Barriers in collecting structured, national data on a frequent basis are discussed, and several recommendations are provided of what is needed to move towards a data ecosystem that can help us to track the development and mental well-being of all children and youth and the impact of the care they receive.

## 1. Introduction

The great burden on care due to mental health problems has become clear in the past few decades. COVID-19 has increased the attention for mental health, as it appeared that several measures to contain COVID-19 did, and still do, have a negative impact on physical and mental health for many young people [1,2]. Compared to the pre-COVID-19 era, at the present time, many adolescents are more likely to suffer from symptoms such as depression, anxiety, and loneliness [3]. This resulted in an increased demand on youth mental health care, and the already long waiting lists grew even longer [4]. Many young people with mental health problems do not receive care, which is not only due to the long waiting lists but also to the stigma surrounding mental health problems, experienced barriers, and difficulties with regard to seeking help [5]. 

Although the Netherlands is gradually beginning to gain insight into the consequences of COVID-19 measures on the mental health of young people, a comprehensive, reliable, and national overview is still far from complete. Even though the Netherlands is one of the richest countries in the world [6], with high-quality youth mental health services [7], it is missing nationally representative data on child and adolescent mental health, e.g., on prevalence, course, and consequences of psychological problems, disorders, and mental health care utilization. Researchers and policy makers also lack a basic data infrastructure that is necessary to provide timely and reliable data crucial for benchmarking and informed decision making. 

In this article, we describe the necessity for a clear and well-organized overview of data on youth mental health and mental health care. We look back on three key moments in time to illustrate the breadth of the desire for data: (1) the transformation of the Dutch youth care system to a decentralized system (2015); (2) the increase in mental pressure and the peak in suicide numbers in youth, which sparked conversation in the media and politics (2018); and (3) the need for data on mental health (and care) during the COVID-19 pandemic (2020–present). Per key moment, we describe the gaps in knowledge and the consequences of lacking a national mental health data infrastructure. We conclude with several recommendations of what is needed to move towards such a data ecosystem that can help us to track the development and mental well-being of all children and youth, and the impact of the care they receive. 

## 2. 2015: Decentralization of Youth Care

On 1 January 2015, the Dutch Child and Youth Act (in Dutch: Jeugdwet) was implemented. The Child and Youth Act replaced the Youth Care Act of 2004 and adopted several parts of other laws such as the Health Insurance Act. Through the Child and Youth Act, responsibilities previously assigned to national and provincial governments were transferred to local municipalities. Municipal responsibility now includes the full range of welfare, support, and care for children and their families. The aim of decentralizing youth care is to simplify the youth system and to make it more efficient and effective, with the ultimate goal of improving the young person’s own strength and improving the caring and problem-solving capacity of his/her social environment [8]. Although the transition of youth services indeed took place at the start of 2015, the actual transformation of youth care continued in the following years and is still going on today. Transitioning youth care from national to local government fit the trend of decentralizing other forms of care, such as the Social Support Act. 

Although the implementation of the Child and Youth Act entailed sufficient necessity to measure the implementation process and its effects, unfortunately, this necessity is not yet widely recognized. As a result, until now, hardly any action has been taken to monitor the roll-out and the effects of this act. Some figures are available, such as the number of children entering youth care before and since the introduction of the Child and Youth Act. These numbers show that in 2000, 1 in 27 children prior to age 18 received a form of youth care, which increased in 2015 to 1 in 9 to 10 [9], with referrals peaking at 1 in 7 to 8 children in 2021 [10]. Seeing these numbers, it appears that the transition of youth services is not reflected in less youth care utilization. Unfortunately, we do not know much more, which makes us unable to show whether the decentralization of youth care has led to other, hopefully more favorable, figures.

One big cause of the difficulty in comparing data (e.g., between different youth care facilities on demographics and diagnoses) before and after implementation of the Child and Youth Act, is the changed registration systems of youth care. Before 2015, youth care was divided over different laws, sectors, and financial systems, which made it difficult, if not impossible, to aggregate data. Transitioning care to a single party, the municipalities, could have made the comparison of data over the years more simplistic and accurate. However, in practice, a myriad of local and regional structures and systems for data collection and monitoring, makes comparison of data even more complicated, and generating a detailed national overview is still practically impossible. The lack of representative data makes it difficult to evaluate if the care provided to children and their families in the decentralized youth care system is efficient and effective, which is one of the original goals of the transformation. Youths themselves have emphasized the importance of the quality of care in several national and international publications, e.g., [11,12]. Without representative data on mental health care utilization and outcome evaluation, ensuring the quality of care is more difficult. 

## 3. 2018–2019: Alarming Signs Regarding Youth Mental Health?

The attention to mental health problems in the Netherlands is not new to the present times of COVID-19, but also existed for quite some time before the pandemic. This focus started with several Dutch research reports that came out in 2018. These reports signaled increased societal expectations and mental pressure among young people [13,14]. In addition, the Health Behaviour in School-aged Children (HBSC) report was published. This study found no increase in mental health problems from 2005 to 2017. However, it was noted that since 2001, the percentage of students experiencing high pressure from schoolwork had doubled and several risk groups for mental health problems were identified (e.g., having a lower education level or not living with both parents) [15]. In July 2018, Statistics Netherlands (CBS) announced that the number of deaths by suicide among youths from age 10 up to (but not including) 20 had risen to 81 in 2017. In previous years, the numbers had always been around 50 and below 60: in 2013 there were 58; in 2014 there were 55; and in both 2015 and 2016, there were 48 suicides among youths from age 10 up to 20. These four reports led to a series of alarming articles in the media during the summer period of 2018.

In order to better understand and interpret these alarming signs, a follow-up study was launched after September 2018. This study was conducted to gain more insights into the available data on mental health problems, mental pressure, and stress among youth and young adults [16]. It was concluded, among other things, that we do not have a clear picture of the mental health of young people. In previous studies, the focus was mostly on psychological complaints, but youth with elevated scores on psychological complaints are a mixed group of individuals with mild symptoms or a disorder. No national figures were available on mental disorders. In addition, other reports that came out around that time indicated a lack of data due to the transfer of youth mental health care to municipalities from 2015. Because of this transfer, there are no longer any nationally available data on clients up to 18 years of age in the Generalist Basic Mental Health Care and Specialist Mental Health Care [17]. The WHO also recently reported the Netherlands as one of the countries, together with other western European countries such as Germany, France, and Great Britain, that is not able to provide estimates on the rate of youth under 18 treated by a mental health professional for ADHD, autism, or depression or the number of prescriptions issued for ADHD, autism, or depression [18]. With regard to the increase in suicides among youth, and the lack of available data in this regard, a multi-method psychological autopsy study was conducted to assess feasibility, identify related factors, and study the interplay of these factors to inform suicide prevention strategies [19]. In this report, the authors made a strong plea for an infrastructure to continuously monitor, evaluate, and support families after each youth suicide and to improve prevention strategies. 

## 4. 2020–2022: The Impact of COVID-19 on Youth Mental Health

A third key moment to illustrate the breadth of the desire for national data is the start of the COVID-19 pandemic, which occurred shortly after the above-mentioned surveys were conducted. COVID-19 clearly emphasized that we lack a basic data infrastructure from which we can easily obtain a complete and national picture of the different facets of mental health, such as the prevalence and course of psychological problems and mental health care utilization. Because we are behind the times, many studies were initiated to provide insight into the consequences of the crisis for youth mental health. In 2021, the Dutch Youth Institute compiled all Dutch studies that had been conducted until then, which was over one hundred studies [20]. It was concluded that youth experienced decreased well-being and more psychological complaints as a result of the crisis. Unfortunately, quality assessment of the included studies did not take place, which means that the only study conducted in the Netherlands also contains qualitatively poor and limited studies, for example, cross-sectional studies in very specific populations and/or with small sample sizes. As a consequence, the results of this overview are not completely reliable. 

In 2022, a literature review was released by the National Institute for Public Health and the Environment (RIVM) and the Netherlands Institute for Health Services Research (NIVEL) that compiled both national and international literature [13]. The international studies within this systematic review covered the period prior to Fall 2020; the Dutch studies also included Spring 2021. The literature review showed that the COVID-19 crisis had a negative impact on physical and mental health for many Dutch young people. Many adolescents were more likely to suffer from symptoms such as depression, anxiety, and loneliness. Among youth who already experienced mental health problems, there were more negative effects from the crisis, and their existing problems became worse. It will be difficult to anticipate the long-term consequences of COVID-19 on the mental health of children and adolescents because there is currently no long-term data. This is an important issue for the coming years. 

It is telling that numerous studies had to be started—quickly and completely unprepared—hoping to obtain a handle on the effects of COVID-19 on youth mental health. Unfortunately, already existing monitoring practices in the domain of well-being were of limited use in monitoring mental health and health care utilization before, during, and after COVID-19. This is mostly due to the fact that the monitoring practices in place have been slow in delivering their rich data: the results of youth monitors are now usually released a year after administration. In a crisis situation, a basic data infrastructure which allows short-cyclical monitoring and reporting may help to obtain a clear picture of developments in mental health.

To summarize, the transition of youth care to municipalities, the peak in suicides in 2018, and the COVID-19 pandemic illustrate and emphasize the need for systematically collected data to evaluate policy and care on a frequent basis, to be able to explain current developments, and to anticipate future unexpected events. Taking a closer look, the Netherlands lacks nationally representative data on the occurrence, course, and consequences of mental disorders in young people up to 18 years of age, including data on (patterns of) health care utilization [16,17].

## 5. Why Do We Need Data?

Every child has the right to protection and should have access to care. Under the Dutch Child and Youth Act, municipalities have been given the responsibility of organizing appropriate care for the youth in their municipalities. In practice, there seems to be a lack of data which leads to a mismatch between the available care and the demand that municipalities and care providers are facing. When figures on the extent and severity of the problems can be placed alongside figures on the requests for help and the care provided, better choices can be made with regard to the organization of the supply of care and purchasing policy.

Second, monitoring the mental health of youth does not only have an advisory function but also a signaling function. Monitoring mental health can function as a quality assessment of care. Such a quality-of-care assessment is needed, whereby we need to assess how children and adolescents enter care, what care is provided, and whether the provided care actually contributed to well-being, recovery, or improved participation. Such a study was already carried out in the Northern Netherlands (C4Youth; an Academic Workplace for Youth in the province of Groningen) before the transition but was not continued after the transition because the need for follow-up measurements was not supported by financial contributions.

Since the start of the Child and Youth Act and the decentralization of youth care, the field has become very fragmented. As an example, youth care in 2019 was provided by more than 6000 providers [21]. There is not a standardized manner in which providers collect data on their population characteristics, utilization, and outcome measures. The available data from these providers is difficult to match, and subsequently, a national overview to gain insight into why young people receive care, what kind of care they receive, and whether the provided care works is absent.

To sum up, there is a general lack of insight into the care processes and the mechanisms that ultimately lead to certain outcomes. This includes data on access, content, continuity, coordination, and completeness of care. In addition, there is little or no insight into the referral processes, treatment decisions, collaborations, and/or consultation structures.

## 6. Barriers and Solutions in Collecting Structured, National Data on a Frequent Basis

In December 2021, several Dutch organizations that have a role in monitoring youth mental health care organized a meeting to discuss the current situation and to think about future directions. During this meeting, in which representatives of national knowledge institutes, universities, youth mental health care, and policy makers were present, including all authors of this paper. It was concluded that even though we have several necessary monitors and collaborating initiatives, a clear overview of youth mental health and associated care is missing.

Several barriers for such an overview were observed:

There are no uniform definitions of mental health or consensus on which aspects mental health includes; see also [22].

The data that we have comes from many different sources, is fragmented, insufficiently connected, and is not easily comparable [16].

There are many different initiatives that are working on or towards the same goals with regard to data collection, but these initiatives do not necessarily align or work together.

There are privacy and practical barriers that hinder the combination of data. Legal and contractual matters, such as General Data Protection Regulation (GDPR), processing agreements, and privacy statements take time and specific expertise.

Data are collected for different purposes and within different systems.

First, we need agreement on definitions and a common language with regard to mental health which will enable comparisons of data across databases [22,23]. Ideally, this will lead to various broadly supported variables and characteristics, which can be collected in the same way across different sectors in the youth field. In the Netherlands, The Trimbos Institute and the National Institute for Public Health and the Environment are currently working on a broadly supported definition of mental health with a Delphi study.

Secondly, ministries and grant agencies could work more closely together to avoid fragmentation and to work towards a sustainable data infrastructure. Current research programs and grants create fragmentation in data collection. The majority of data collection initiatives are focused on the short term, often related to the temporality of the project within which the data are collected. Data on long-term outcomes (for example, what are the outcomes of youth mental health care at a later age) are virtually absent. It is necessary to see what connections are possible between different data sets and monitors on youth mental health and mental health care. Where is the overlap? And for which questions can existing data be used? To enable this, national resources are needed to take stock of what data already exist, to harmonize these data where possible and needed, and to bring the different data initiatives together.

Thirdly, we propose the development of a data ecosystem for tracking the development and mental well-being of all children and youth and to track the impact that it has on youth mental health care. Much data collection is organized locally and often lacks the opportunity to provide a national overview of processes and outcomes. As a result, the potential for mutual learning is insufficiently exploited. At the moment, each initiative has to take care of certain preconditions, such as satisfying the GDPR (General Data Protection Regulation) requirements, for example. It is more efficient to have experts arrange this centrally and to set up a central infrastructure for these kinds of issues. Thus, to enable working in the youth welfare sector in a scientifically responsible way and to establish effective policy, a robust knowledge infrastructure is needed, including a well-functioning national digital mental health data infrastructure. In doing so, we advise using measurement tools and monitoring systems that uniformly and reliably capture care processes and psychiatric epidemiological outcome measures. This ecosystem should also allow local care processes and outcomes to be adequately compared with national care processes.

## 7. Conclusions

To conclude, there seems to be an increased sense of urgency with regard to adequately monitoring youth mental health and associated care. UNICEF Netherlands and the National Children’s Ombudsman advocate for better data and monitoring systems with regard to child and adolescent mental health. Monitoring is also a topic in the intended youth care reforms and in the intended National Approach on Mental Health Promotion of the Dutch Ministry of Health. We hope that the advice and investments as described above will be included in these developments and that we can start working together on a sustainable mental health data ecosystem.

## Data Availability

Not applicable.

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
