# Peer review of "The Need for a Data Ecosystem for Youth Mental Health in The Netherlands"

_ijerph, 2022, doi:10.3390/ijerph191811499_

Round 1

Reviewer 1 Report

This is a policy oriented communication focusing on youth mental health care  in the Netherlands. The timing is good, and the topic highly relevant. However the way it is presented could be improved. The paper lacks the context eg the Netherlands is a rich country with very high quality youth mental health services; some of the issues presented will also be present in other countries and in adult mental health care services. The voice of young people is lacking. Also, the fact that most young people with mental health problems do not receive care is not mentioned, including the fact that many young persons are scared to ask for help is. Some of the statements made by the author, needs to be nuanced a bit. Eg we do know quite a bit about barriers in access to care. The cited references are mainly Dutch from policy institutions, and literature from clinicians who deal with young people on a daily basis is lacking. I wonder whether this paper is more suitable for a Dutch journal, given that it seems only relevant for the Dutch situation?

Author Response

Reviewer 1

  1. This is a policy oriented communication focusing on youth mental health care  in the Netherlands. The timing is good, and the topic highly relevant. However the way it is presented could be improved.

RE: We thank the reviewer for the compliments regarding our paper.

  1. The paper lacks the context eg the Netherlands is a rich country with very high quality youth mental health services; some of the issues presented will also be present in other countries and in adult mental health care services.

RE: We agree with the reviewer that more context could be added. On page 1, in paragraph 2, we added:

“Even though the Netherlands is one of the richest countries in the world [5] with high quality youth mental health services [6], it is missing nationally representative data on child and adolescent mental health, e.g., on prevalence, course and consequences of psychological problems, disorders and mental health care utilization.”

And on page 3:

The WHO also recently reported The Netherlands as one of the countries that is not able to provide estimates on the rate of under-18s treated by a mental health professional for ADHD, autism and depression or the number of prescriptions issued for ADHD, autism and depression, together with other western European countries such as Germany, France, and Great Britain [17]”

With regard to issues present in adult mental health care we decided not to add this to the manuscript. The focus of the present manuscript is on youth mental health care. Going into the differences and similarities with adult mental health care would warrant an additional paragraph and take up much space. It would in our opinion distract from the focus and main messages of the paper.

  1. The voice of young people is lacking.

RE: The reviewer is right that we did not include the voice of young people in our manuscript. Based on the comment of the reviewer we conducted an extensive search into national and international research on the standpoint of young people with regard to the importance of reliable data on mental health. We were not able to retreive any reports that specifically investigated this topic. There are however reports that indicate that youth find the quality of mental health care important. The quality is not well monitored however. We therefore added the following on page 2:

“Youth themselves have emphasized the importance of the quality of care in several national and international publications [e.g. 11, 12]. Without representative data on mental health care utilization and outcome evaluation, ensuring the quality of care is more difficult.” 

  1. Also, the fact that most young people with mental health problems do not receive care is not mentioned, including the fact that many young persons are scared to ask for help is.

RE: we added this information on page 1, paragraph 1:

“Many young people with mental health problems do not receive care, which is not only due to the long waiting lists, but also to stigma surrounding mental health problems and experienced barriers and difficulties with regard to seeking help [5].”

  1. Some of the statements made by the author, needs to be nuanced a bit. Eg we do know quite a bit about barriers in access to care.

RE: We removed the words ‘(barriers to)’ from the first paragraph on page 5.

  1. The cited references are mainly Dutch from policy institutions, and literature from clinicians who deal with young people on a daily basis is lacking.

Based on the reviewers comments we added 7 references, 5 of which are peer-reviewed international articles.

  1. I wonder whether this paper is more suitable for a Dutch journal, given that it seems only relevant for the Dutch situation?

RE: The paper is especially written for a special issue focusing on the care and health of children in the Netherlands: “The Health of Children in the Netherlands: State of the Art, Challenges Ahead and Perspectives for Future Research.” In addition, this paper can certainly be of interest to a wider, international audience because - as you rightly pointed out earlier - the problem is unfortunately not unique to the Netherlands.

Reviewer 2 Report

Thank you for the opportunity to review this paper. This manuscript is a potentially important publication which highlights an issue of significant ongoing concern in youth mental health: poor data collection and availability. A particular strength of this manuscript is the authors' approach to representing opportunities in time when data could have had a meaningful impact on progress, and how the lack of data is effecting that. Additionally, the authors' suggestions are sensible and logical given the data landscape they portray. Overall, I feel the merits of this paper are strong. I have a few significant suggestions which I feel could strengthen the paper overall: 

- The 'Introduction' section focuses on Covid, with some vague references to the negative impact of measures to contain Covid on mental health. Additional citations would strengthen this argument (I'm curious if anyone has specifically identified that containment measures effected mental health, and how that research was conducted controlling for other factors!). Additionally, as the COVID-19 pandemic is only one of the three situations discussed in this manuscript, I feel this introduction could be strengthened with greater references to the broader data context for youth mental health both within The Netherlands and abroad. 

- In section 4, greater detail would benefit the authors' argument about why the quality assessment indicated data quality was poor. What are the issues with the current data that make it unreliable?

- Similarly, in line 152 the authors discuss existing monitoring practices that are of limited use. What are these practices and why is their utility limited?

- In line 185, the authors discuss the high numbers of care providers and note that it is therefore undesirable to lack a national overview of care. I think some greater discussion is needed to make this link more explicit. Why is this particularly concerning given the large number of providers? Would this be less concerning if there were fewer providers? 

Minor suggestions

- The use of colloquial language throughout the manuscript should be addressed. Examples include 'coming from a broken home' (line 99) 'get a grip on the effects' (line 151).

- Similarly, when discussing suicide statistics please be specific as to what you are talking about e.g., 'number of deaths by suicide' instead of 'number of suicides' (line 100). 

- The use of the phrase 'peak in suicides' (e.g., in line 158) suggests that youth suicide rates were highest in 2018 and have subsequently declined. Have youth suicide rates remained high in The Netherlands? In that case, I would recommend describing this as an 'increase' rather than a 'peak'. 

Author Response

Reviewer 2

  1. Thank you for the opportunity to review this paper. This manuscript is a potentially important publication which highlights an issue of significant ongoing concern in youth mental health: poor data collection and availability. A particular strength of this manuscript is the authors' approach to representing opportunities in time when data could have had a meaningful impact on progress, and how the lack of data is effecting that. Additionally, the authors' suggestions are sensible and logical given the data landscape they portray. Overall, I feel the merits of this paper are strong. I have a few significant suggestions which I feel could strengthen the paper overall.

RE: We thank the reviewer for these positive words about our manuscript.  

  1. The 'Introduction' section focuses on Covid, with some vague references to the negative impact of measures to contain Covid on mental health. Additional citations would strengthen this argument (I'm curious if anyone has specifically identified that containment measures effected mental health, and how that research was conducted controlling for other factors!). Additionally, as the COVID-19 pandemic is only one of the three situations discussed in this manuscript, I feel this introduction could be strengthened with greater references to the broader data context for youth mental health both within The Netherlands and abroad. 

RE: We agree with the reviewer that providing a broader context would improve the manuscript. Also, we acknowledge the difficulties relating the covid measures to specific mental and physical health outcomes. However, although an exacerbation of mental health problems cannot be attributed 1 to 1 to COVID-19, there are studies that can clearly show a trend by, for example, using the same design and methodology before and during COVID-19. We included two such studies and rewrote the introduction paragraphs as follows, adding five additional references in total:

“The great burden on care due to mental health problems has become clear in the past few decades. Covid-19 has increased the attention for mental health, as it appeared that several measures to contain Covid-19 did and still have a negative impact on physical and mental health for many young people [1, 2]. Compared to the pre-Covid-19 era, nowadays many adolescents are more likely to suffer from symptoms such as de-pression, anxiety, and loneliness [3]. This resulted, among others, in an increased de-mand on youth mental health care, and the already long waiting lists grew even longer [4]. Many young people with mental health problems do not receive care, which is not only due to the long waiting lists, but also to stigma surrounding mental health problems and experienced barriers and difficulties with regard to seeking help [5].

Although the Netherlands is gradually beginning to gain insight into the consequences of Covid-19 measures on the mental health of young people, a comprehensive, reliable, and national overview is still far from complete. Even though the Netherlands is one of the richest countries in the world [6] with high quality youth mental health services [7], it is missing nationally representative data on child and adolescent mental health, e.g., on prevalence, course and consequences of psychological problems, disorders and mental health care utilization. Researchers and policy makers also lack a basic data infrastructure that is necessary to provide timely and reliable data crucial for benchmarking and informed decision making.”

Added references:

  1. Viner, R.; Russell, S.; Saulle, R.; Croker, H.; Stansfield, C.; Packer, J.; ... & Minozzi, S. School closures during social lockdown and mental health, health behaviors, and well-being among children and adolescents during the first COVID-19 wave: a systematic review. JAMA pediatrics, 2022, 176(4), 400-409.
  2. Ravens-Sieberer, U.; Kaman, A.; Erhart, M.; Devine, J.; Schlack, R.; Otto, C. Impact of the COVID-19 pandemic on quality of life and mental health in children and adolescents in Germany. Eur Child Adolesc Psychiatry. 2022, 31(6), 879-889.
  3. Tuijnman, A.; Kleinjan, M.; Olthof, M.; Hoogendoorn, E.; Granic, I.; & Engels, R. C. M. E. A Game-Based School Program for Mental Health Literacy and Stigma on Depression (Moving Stories): Cluster Randomized Controlled Trial. JMIR Mental Health, 2022, 9(8), e26615.
  4. Global Finance Magazine. Richest Countries in the World 2022. Act. Retrieved 26 Aug 2022 from https://www.gfmag.com/global-data/economic-data/richest-countries-in-the-world.

2.7.       Kilbourne, A. M.; Beck, K.; Spaeth‐Rublee, B.; Ramanuj, P.; O'Brien, R. W.; Tomoyasu, N.; Pincus, H. A. Measuring and improving the quality of mental health care: a global perspective. World Psychiatry, 2018, 17(1), 30-38.

 In addition, on page 3 we added:

The WHO also recently reported The Netherlands as one of the countries that is not able to provide estimates on the rate of under-18s treated by a mental health professional for ADHD, autism and depression or the number of prescriptions issued for ADHD, autism and depression, together with other western European countries such as Germany, France, and Great Britain [11]”

We would like to emphasize that the paper is especially written for a special issue focusing on the Netherlands: The Health of Children in the Netherlands: State of the Art, Challenges Ahead and Perspectives for Future Research.

  1. In section 4, greater detail would benefit the authors' argument about why the quality assessment indicated data quality was poor. What are the issues with the current data that make it unreliable?

We added the following examples of qualitatively poor and limited studies to section 4:

“Unfortunately, quality assessment of the included studies did not take place which means that the only study conducted in the Netherlands also contains the qualitatively poor and limited studies, for example cross-sectional studies in very specific populations and/or with small sample sizes.”

  1. Similarly, in line 152 the authors discuss existing monitoring practices that are of limited use. What are these practices and why is their utility limited?

We realize that we could have been more clear. The limited use of the existing monitoring practices is described in the sentence after this statement. It was formulated as follows:

“It is telling that numerous studies had to be started – quickly and completely unprepared – hoping to get a grip on the effects of Covid-19 on youth mental health. Un-fortunately, already existing monitoring practices in the domain of well-being were of limited use in monitoring mental health and health care utilization before, during and after covid. Until now, the monitoring practices in place have been slow in delivering their rich data: the results of youth monitors are now usually released a year after administration. In a crisis situation, a basic data-infrastructure which allows short-cyclical monitoring and reporting may help to obtain a clear picture of developments in mental health.”

We have rewritten this parapgraph as follows on page 4:

“It is telling that numerous studies had to be started – quickly and completely unprepared – hoping to get a grip on the effects of Covid-19 on youth mental health. Unfortunately, already existing monitoring practices in the domain of well-being were of limited use in monitoring mental health and health care utilization before, during and after covid. This is mostly due to the fact that the monitoring practices in place have been slow in delivering their rich data: the results of youth monitors are now usually released a year after administration. In a crisis situation, a basic data-infrastructure which allows short-cyclical monitoring and reporting may help to obtain a clear picture of developments in mental health.” 

  1. In line 185, the authors discuss the high numbers of care providers and note that it is therefore undesirable to lack a national overview of care. I think some greater discussion is needed to make this link more explicit. Why is this particularly concerning given the large number of providers? Would this be less concerning if there were fewer providers? 

We realize that this is indeed unclear. We changed this passage as follows on page 4:

“Since the start of the Child and Youth Act and the decentralization of youth care, the field has become very fragmented. As an example, youth care in 2019 was provided by more than 6,000 providers [16]. There is not a standardized manner in which providers collect data on their populations characteristics, utilization and outcome measures. The available data from these providers is difficult to match and subsequently we do not have a national overview to gain insight into why young people receive care, what kind of care they receive and whether the provided care works is absent.”

Minor suggestions

  1. The use of colloquial language throughout the manuscript should be addressed. Examples include 'coming from a broken home' (line 99) 'get a grip on the effects' (line 151).

RE: We corrected this. ‘Coming from a broken home’ was changed into ‘not living with both parents’. ‘Get a grip on the effects’ was changed into ‘obtain a handle on the effects’

  1. Similarly, when discussing suicide statistics please be specific as to what you are talking about e.g., 'number of deaths by suicide' instead of 'number of suicides' (line 100). 

RE: We changed this accordingly.

  1. The use of the phrase 'peak in suicides' (e.g., in line 158) suggests that youth suicide rates were highest in 2018 and have subsequently declined. Have youth suicide rates remained high in The Netherlands? In that case, I would recommend describing this as an 'increase' rather than a 'peak'. 

RE: It is indeed the case that youth suicide rates were highest in 2018 and have subsequently declined.